# Incorporating Graph Attention and Recurrent Architectures for City-Wide Taxi Demand Prediction

**Ying Xu [1] and Dongsheng Li [1,*]**

National Lab for Parallel and Distributed Processing (PDL), School of Computer Science, National University of Defense Technology, Changsha 410073, Hunan, China; xuying08@nudt.edu.cn

*   Correspondence: dsli@nudt.edu.cn

**Abstract:** Taxi demand prediction is one of the key factors in making online taxi hailing services more successful and more popular. Accurate taxi demand prediction can bring various advantages including, but not limited to, enhancing user experience, increasing taxi utilization, and optimizing traffic efficiency. However, the task is challenging because of complex spatial and temporal dependencies of taxi demand. In addition, relationships between non-adjacent regions are also critical for accurate taxi demand prediction, whereas they are largely ignored by existing approaches. To this end, we propose a novel graph and time-series learning model for city-wide taxi demand prediction in this paper. It has two main building blocks, the first one utilize a graph network with attention mechanism to effectively learn spatial dependencies of taxi demand in a broader perspective of the entire city, and the output at each time interval is then transferred to the second block. In the graph network, the edge is defined by an Origin–Destination relation to capture non-adjacent impacts. The second one uses a neural network which is adept with processing sequence data to capture the temporal correlations of city-wide taxi demand. Using a large, real-world dataset and three metrics, we conduct an extensive experimental study and find that our model outperforms state-of-the-art baselines by 9.3% in terms of the root-mean-square error.

**Keywords:** taxi demand prediction; Origin–Destination; graph; attention; GRU

---

## 1. Introduction

Online taxi hailing services such as Uber, Lyft, and DiDi Chuxing have become an important part of today's intelligent transportation system (ITS). The success of online taxi hailing depends on many factors, among which, demand prediction, i.e., forecasting the number of rides in a specific area during the next period of time, is one of the most critical.

Accurate city-wide taxi demand prediction can lead to tremendous benefits. For instance, it allows service providers to preallocate taxis to highly-demanded areas, reducing both the waiting time of passengers and the idle time of taxis. Meanwhile, increased taxi utilization could raise drivers' incomes and reduce energy waste. From a broader point of view, taxi demand prediction also helps to optimize traffic efficiency via alleviating the imbalance of transport capacity across the city. It is worth noting that there is a series of works studying Autonomous Mobility on-Demand (AMoD) systems which also aims at alleviating the disequilibrium of taxi demands among areas [1–5]. However, we have a different task from them. In AMoD, the task to dispatch ride-sharing vehicles to meet taxi demands, while in our work, the task is to predict taxi demands.

Due to its great practical value, taxi demand prediction has been extensively investigated. Existing solutions can be roughly classified into two categories: model-based and deep learning-based. Model-based approaches assume that taxi demand follows certain predefined patterns. Explored models/patterns include the AutoRegressive Integrated Moving Average (ARIMA) [6,7],

Gaussian mixture [8], and linear regression [9], to name a few. Model-based methods are simple and easily implementable, but suffer from not having ability to fully capture the complex spatial and temporal dependencies of taxi demand. In contrast, deep learning models can offer more powerful expressiveness. As taxi demand prediction is essentially a time series processing problem, Recurrent Neural Networks (RNNs), which are skilled in sequence processing, is effective for the task [10]. In addition, it has become a popular trend to exploit RNNs in conjunction with the Convolutional Neural Networks (CNNs) for demand prediction [11–13], owing to the ability of CNNs to mine spatial correlations. On the other hand, efforts are also made to solely leverage CNNs or its variants [14,15].

Despite these achievements, existing deep learning approaches still have the following limitations:

- They only consider the spatial influences between adjacent areas, as demonstrated in Figure 1a, while the influence between non-adjacent areas is rarely considered. However, the latter spatial influence is also crucial for city-wide taxi demand prediction. For instance, the demands at two railway stations in the same city are very likely to be correlated, even though the two stations are far away from each other.

- In CNNs models, the impacts of adjacent areas (those blue in Figure 1a) are fixed, which limits the flexibility of learning diverse spatial patterns of various areas to a large extent. Zhou et al. [16] consider self-adaptive weight assignment with attention, however, they only focused on a fixed set of representative areas, instead of all possible areas of the city, for different target areas.

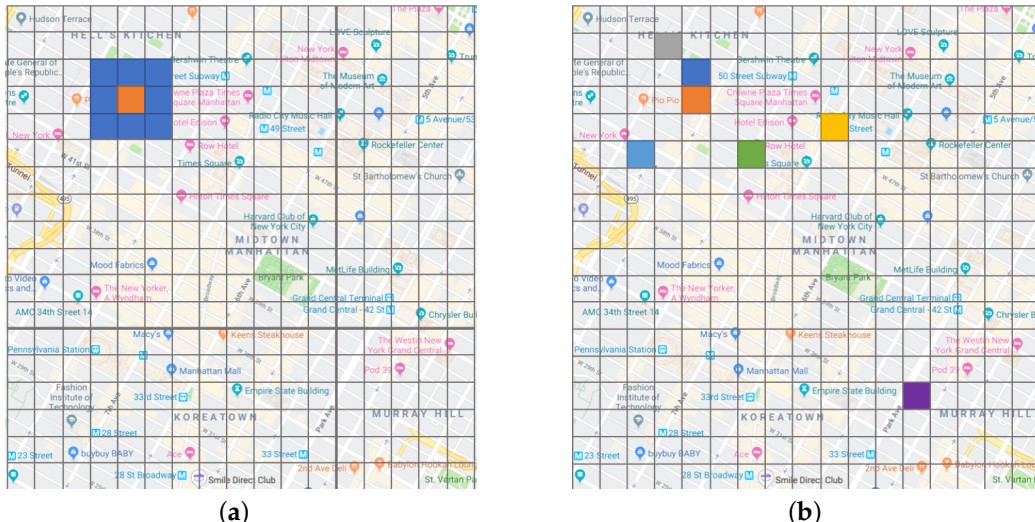

(**a**)                   (**b**)

**Figure 1.** Neighbors Comparison between the existing CNNs-based methods and the Graph Networks based method used in our work. (**a**) The center grid (the orange one) and its neighboring grid cells (blue ones) in the previous CNNs-based methods. The city is regarded as an image. In this image, all neighboring grid cells are adjacent to the center grid and transfer the same influence to the center grid. (**b**) The center grid (orange one) and its neighbor grids (other colored ones) in our approach. We consider the city as a graph and each grid as a node. The neighbors in the graph may not be adjacent to the center grid in the physical world, and the neighbors could have different impacts on the center grid.

To solve the above issues, we propose a framework based on graph networks. We first partition the entire city into a set of grids and then transform grids into a graph, each grid is a node in the graph. The edges between nodes are defined by an Origin–Destination (OD) relationship. If a taxi picks up passengers in grid *A* and drops off passengers in grid *B*, then grid *A* (Origin) and grid *B* (Destination) form a pair with the OD relationship, thus there is an edge between the corresponding nodes in the graph. Since the graph has a non-Euclidean structure which is unable to be handled by CNNs [17,18], we employ a graph attention network, which incorporates an attention mechanism

into the Graph Convolutional Networks (GCN). The GCN is inspired by spectral theory and extracts features of nodes by leveraging the eigenvalues and eigenvectors of the Laplacian matrix of the graph. Moreover, considering that, for a grid, the neighbors adjacent to it have different influences on it, we leverage the attention mechanism to assign weights to the edges. In addition, when mining data features, we not only make full use of external information such as time and space coordinates, but also operate on the historical data to obtain richer features by using average models or carrying out a Fourier transformation. Our paper makes notable contributions summarized as follows:

- To the best of our knowledge, we are the first to use the graph networks to deal with the taxi demand prediction problem. Different from the previous methods, we model the city as a graph and exploit the influences between non-adjacent areas sufficiently by defining the edges via OD relations in the graph. Note that the concept of OD is also used by Hamedmoghadam et al. [19], but they aim at predict flow between ODs, while we regard OD as an assistant to improve taxi demand prediction.
- We are conscious of the different impacts from multiple neighbors for each center node. Thus, the graph network with attention mechanism is employed to capture the difference. We propose the Origin–Destination-based Temporal Graph Attention Network (OD-TGAT), in which Graph Attention Networks (GAT) is employed to capture spatial relationships with different edge weights.

The remainder of this paper is organized as follows. Section 2 reviews the related work. Some preliminaries are given in Section 3. We describe our methods in Section 4. The experimental results and analysis are presented in Section 5. We conclude this paper in Section 6.

## 2. Related Work

### 2.1. Taxi Demand Prediction

Initially, methods used to solve taxi demand prediction were generally traditional ones such as statistical approaches. As deep learning is becoming popular, more attention has turned to deep learning models, but the pace of exploration using traditional methods has not stopped. The following two parts introduce the evolution of these two methods.

2.1.1. Traditional Methods

Solving the problem of taxi demand forecasting by traditional methods has a long history and it is still enduring. As taxi demand data are one kind of time series data, it is common to use time series analysis approaches to handle this task. Sun et al. [20] use exponential smoothing, a well-known forecasting method in time series analysis, to capture the longer-term trends and essential laws behind the data. Li et al. [6] incorporate the representative time series prediction method ARIMA into the model to forecast the number of passengers at the next time interval. Besides time series analysis methods, clustering based approaches are also widely used. Chiang et al. [8] employ Poisson process and the Gaussian Mixture Model (GMM) to cluster the spatio-temporal points, predicted demands is sampled from clusters. Zhang et al. [21] utilize the adaptive Density-Based Spatial Clustering of Applications with Noise (DBSCAN) to predict taxi demand hotspots. Davis et al. [22] adopt the geohash technique and develop a multi-level clustering method to improve the accuracy of linear time-series model. Li et al. [23] propose an adaptive clustering algorithm and a novel regressor to predict how many bikes are to be rented and returned throughout the city. The third type of method to operate taxi demand prediction is regression. Tong et al. [9] use the linear regression model for forecasting with notable feature engineering. Qian et al. [24] use the geometrically weighted regression (GWR) to model the spatial heterogeneity of the taxi ridership. In addition to the methods mentioned above, there have also been some other trials. For example, the representative neural network in machine learning, multilayer perceptron (MLP), is used by Mukai et al. [25] into their work to analyze

and predict taxi demands. Zhou et al. [26] even employ three predictors to adapt to areas with different resolutions.

Unfortunately, although traditional methods are diverse, they are not able to capture the nonlinear relationships between data in most scenarios, and the defect is made up well by deep-learning-based approaches.

### 2.1.2. Deep-Learning-Based Methods

The widely used deep learning model has greatly accelerated the research progress in various fields, especially in the fields of computer version [27,28] and natural language processing (NLP) [29,30]. Inspired by this, some mainstream deep learning models are also applied to taxi demand prediction. RNNs are famous of predicting time series, so they are the first choice for taxi demand forecast. For example, the Long Short Term Memory (LSTM) networks, a variant of RNNs, are incorporated into the work of [10,31]. LSTM is used in [16] along with the attention mechanism in an encoder-decoder framework to predict multi-step citywide passenger demands. ConvLSTM combined convolutional layer and LSTM is popular too. In [32], ConvLSTM is utilized to address the spatial dependencies and temporal dependencies. In [11], LSTM, ConvLSTM, and convolutional layers are leveraged at the same time. Convolutional operation not only works with ConvLSTM, but also performs well as an individual convolutional layer. For example, Rodrigues et al. [33] integrate word embeddings, convolutional layers, and the attention mechanism for passenger demand prediction. In [12], CNNs are applied at the spatial view, one of the proposed three views. Similarly, the framework in [13] also had three components, CNNs are used in the spatial component as well. Moreover, variants of CNNs could also capture the temporal and spatial features at the same time, such as the dilated causal convolutions used in [15], ResNet adopted in [14]. In addition, Deep Neural Networks (DNN) can also provide a strong learning ability owing to their multiple hidden layers and have shown good performance in passenger demand prediction [34].

Deep-learning-based models have contributed a lot to taxi demand prediction. However, existing works did not take advantage of the power of the city as a whole and did not determine the correlations between pairs of regions at variable distances. Our model is designed to eliminate these issues.

### *2.2. Graph-Convolutional-Based Networks*

The GCN is an approach for handling non-Euclidean structural data that CNNs fail to process well. GCN uses spectral-theory-based graph convolution to efficiently extract spatial features within the graph topology for further learning.

The research of learning models on graphs can be traced back to graph signal processing (GSP) [35,36], which proposed several definitions about the representations of graph signals. Starting with GSP, studies about graph signals gradually divided into two categories: those based on spectral graph theory [37] and those based on graph wavelet theory [38]. The former has become more and more popular since Bruna et al. [39] proposed the GCN, which includes a defined Fourier transform graph based on the Laplacian graph. In terms to huge computations in GCN, a series of works contributed to the acceleration of GCN [40–44] via a series mathematical transformations such as Lanczos method, Chebyshev polynomial filters. In addition, Huang et al. [45] improved the FastGCN [43] by leveraging an adaptive node sampling approach. Notably, as one of the most impactful mechanisms in deep learning, attention mechanisms [46,47] have also been introduced into GCNs. Velivckovic et al. [48] computed the normalized attention on different neighbors for each center node to improve the performance of GCN in classification tasks. Unlike the traditional multi-head attention mechanism, which is also used by [48], Zhang et al. [49] adopted a convolutional sub-network to control the importance of each head's attention. Thekumparampil et al. [50] found, surprisingly, that when they replaced all fully-connected layers in deep networks by a simple linear model, they had comparable achievements with other state-of-the-art approaches.

Our model is inspired by the graph attention network (GAT) [48]. The initial GAT is adaptive to semi-supervised problems. Therefore, in taxi demand prediction, the graph nodes and edges have to be re-defined to apply them to the supervised regression scenario.

## 3. Preliminary

In this section, we first clarify some basic notations, define the taxi demand prediction problem formally, and then briefly explain some principles about GCN.

### 3.1. Definitions

For clarity, Table 1 lists the notations used in this paper.

**Table 1.** Symbols and notations.

| | |
|---|---|
| $\zeta$ | Side length of each grid |
| $M(\zeta)$ | Grid map, the set of grids after the city is partitioned |
| $lat_{\max}, lat_{\min}$ | The maximum latitude, minimum latitude of the city |
| $lon_{\max}, lon_{\min}$ | The maximum longitude, minimum longitude of the city |
| $H, W$ | Number of grids at the longitude, latitude dimension of $M(\zeta)$ separately |
| $g(i, j)$ | The grid of the $i$th along the longitude and of the $j$th along the latitude |
| $s_t, e_t$ | Start and end time of the $t$th time interval separately |
| $\phi$ | Length of each time interval, the unit is minutes |
| $I_t$ | The $t$th time interval |
| $t$ | Abbreviation of $I_t$ in the rest of the paper after defining $I_t$ |
| $\mathcal{G}_t = (V, A)$ | Graph of the city at the $t$th time interval |
| $N_t$ | Number of nodes in $\mathcal{G}_t$ |
| $v_m, v_n, ...$ | Nodes in $\mathcal{G}_t$, belong to $V$ |
| $D_t^{v_n}$ | Taxi demand of node $v_n$ at the $t$th time interval |
| $C_t^{v_n}$ | External features of node $v_n$ at the $t$th time interval |
| $s$ | Time step, the number of historical time intervals for once prediction |
| $\mathcal{D}_{t+1}^{\text{city}}$ | Taxi demand of all nodes in $\mathcal{G}_{t+1}$ |

#### 3.1.1. Region Partition

The city is always divided into a number of areas to play predictions on areas. Dividing methods are various, such as according to the road network [51] or the zip code tabular [52]. We partition the city into a series of grids along the longitude and latitude dimensions, as the *grid_map* illustrated in the left subplot of Figure 2, because this partition method is simple and flexible.

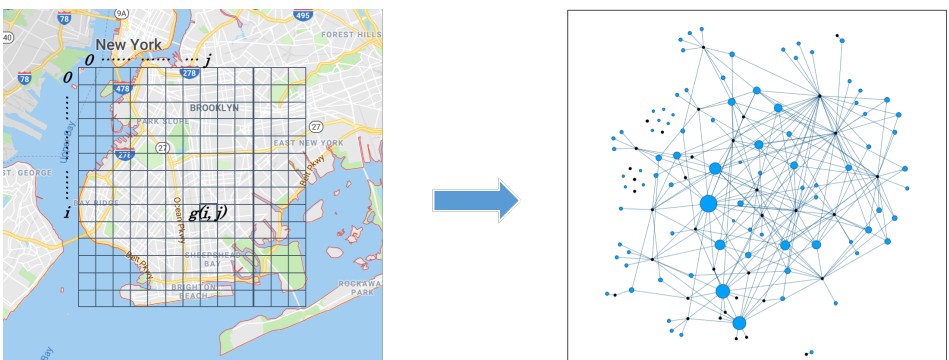

**Figure 2.** Illustration of the region partitioning on a city and the transformed city graph. We use Brooklyn, New York as an example, in which the grid map is obtained after the city being divided along the longitude and latitude, like the left subplot. Afterwards, the grid map is converted into a city graph with grids as nodes and OD relations as edges, like the right subplot.

Assuming the minimum longitude and latitude of the city are $lon_{min}$ and $lat_{min}$, and the maximum values are $lon_{max}$ and $lat_{max}$. Given a grid side length $\zeta$, we partition the city into $H \times W$ grids along the two coordinates axes, where $H = \left\lceil \frac{lat_{max} - lat_{min}}{\zeta} \right\rceil$, and $W = \left\lceil \frac{lon_{max} - lon_{min}}{\zeta} \right\rceil$. The grid of the $i$th along the longitude and $j$th along the latitude is represented by $g(i, j)$ $(i, j \in \mathbb{N}, i \in [0, H), j \in [0, W))$, and the whole *grid map* is represented by $M(\zeta)$.

### 3.1.2. Time Partition

Similarly, we divide time period into time intervals with equal length $\phi$ minutes. The $t$th time interval is expressed by $I_t = [s_t, e_t)$, where $s_t$ and $e_t$ are the start time and the end time of $I_t$ respectively, and they satisfy to $e_t - s_t = \phi$. As a consequence, the number of time intervals in one day is $\frac{1440}{\phi}$. For simplicity, we use the index $t$ to represent time interval $I_t$ for the rest of the paper.

### 3.1.3. City Graph

Given a grid map $M(\zeta)$, the *city graph* is defined as $\mathcal{G}_t = (V, A)$, where $V = \{v_0, v_1, ..., v_N\}$ is the node set of the graph, in which each node corresponds to each grid in $M(\zeta)$ at time interval $t$, as shown in Figure 2. It is worth noting that nodes in $V$ are no more than grids in $M$, i.e., $N \leq H \times W$. The reason for this is that if a grid is not found to have a pickup or dropoff record in it, we regard it as meaningless, and it does not need to be mapped to a node in the graph.

$A \in \mathbb{R}^{N \times N}$ is the adjacent matrix of $\mathcal{G}_t$, which is used to save the weights of edges between nodes. As mentioned in the Introduction section, the edge is defined by OD relations. In other words, if a taxi picks up a group of passengers in $v_m \in V$ and drops them off in $v_n \in V$, there is an edge connecting $v_m$ and $v_n$.

### 3.1.4. Taxi Demand

Based on the notations above, we can define the *taxi demand* on the *city graph*. Given the *grid map* $M(\zeta)$ and time interval $t$(same as $I_t = [s_t, e_t)$), the corresponding *city graph* is $\mathcal{G}_t = (V, A)$. For node $v_n \in V$, whose counterpart grid in $M(\zeta)$ is $g(i, j)$, the *taxi demand* at node $v_n$ at $t$ is defined as $D_t^{v_n}$:

$$
\begin{aligned}
SET_t^{v_n} &= \{(loc, mt) | loc \in g(i, j) \wedge mt \in I_t\} \\
D_t^{v_n} &= |SET_t^{v_n}|
\end{aligned}
\tag{1}
$$

where $loc$ and $mt$ are the location and the moment when a taxi pickup event occurs, $SET_t^{v_n}$ is the set of pickup events, the symbol $|\cdot|$ denotes the cardinality of the set.

### 3.1.5. Citywide Taxi Demand Prediction

When we try to define this problem, we take the city as an entirety into account. The goal of the *citywide taxi demand prediction* problem in this paper is to predict the *taxi demand* of the next time interval for all nodes in the city graph. Given the city graph $\mathcal{G}_t = (V, A)$, $V = \{v_0, v_1, ..., v_N\}$, the current time interval $t$, *time_step $s$*, historical taxi demand data $D$, and additional context features $C$ such as temporal features and spatial features (more details are presented in the Data Description section), the citywide taxi demand of $t + 1$ is

$$
\begin{aligned}
\mathcal{D}_{t+1}^{city} &= \{D_{t+1}^{v_0}, D_{t+1}^{v_1}, ..., D_{t+1}^{v_N}\} \\
&= \{\mathcal{F}(D_{t-s,...,t}^{v_0}, C_{t-s,...,t}^{v_0}), \mathcal{F}(D_{t-s,...,t}^{v_1}, C_{t-s,...,t}^{v_1}), ..., \mathcal{F}(D_{t-s,...,t}^{v_N}, C_{t-s,...,t}^{v_N})\}
\end{aligned}
\tag{2}
$$

where the prediction function $\mathcal{F}(\cdot)$ is shared by all nodes, *time_step $s$* denotes the number of historical time intervals, features of these time intervals are used to predict the demands of the next time interval. Both $D$ and $C$ are limited to be extracted only in $s$ time intervals for prediction, so $D_{t-s,...,t}^{v_n} = \{D_{t-s}^{v_n}, D_{t-s+1}^{v_n}, ..., D_t^{v_n}\}$, and $C_{t-s,...,t}^{v_n} = \{C_{t-s}^{v_n}, C_{t-s+1}^{v_n}, ..., C_t^{v_n}\}$.

## 3.2. Graph Convolutional Networks

Convolutional operations in CNNs is not applicable to general graphs, because the number of adjacent vertices of each vertex in the graph may be different, and the convolutional kernel with fixed size cannot be shared by all vertices. There are currently two classes of methods exploring how to generalize CNNs to structured data forms. One is to expand the spatial definition of a convolution [53], which rearranges the vertices into certain grid forms which can be processed by normal convolutional operations. The other one, called the spectral graph convolution, is to operate in the spectral domain with graph Fourier transforms [39], which introduces the spectral framework to apply convolutions in spectral domains.

The spatial based graph convolution suffers from inefficient since all nodes in the graph have to do all computations, and its performance may not be well as the spectral graph convolution. Therefore, in this work we select the latter one to adapt to our situation.

## 4. Methodology

In this section, we describe the proposed methodology. First we introduce feature engineering, including a description of the dataset and a series of operations on the dataset to obtain features. Afterwards, we demonstrate our model, Origin–Destination-based Temporal Graph Attention Networks (OD-TGAT), including an introduction of the overall framework and details of each component.

## 4.1. Feature Engineering

In this subsection, we introduce the source of our dataset and explain how to clean the raw data, then, we conduct an in-depth analysis of the dataset and obtain a number of rich features.

### 4.1.1. Data Description

New York City (NYC) taxi datasets are provided by the New York City Taxi and Limousine Commission (TLC) [54] and are open to everyone. There are three types of taxi data on the platform: yellow taxis, green taxis, and For-Hire Vehicles (FHVs). Both yellow taxi and green taxi could be hailed by signaling to a driver at the roadside or using an e-hail app. The difference between them is that yellow taxi has longer history while green taxi provides better service. FHV is a taxi that only accepts pre-arranged reservations. As dataset is independent to the proposed model, we choose yellow taxi data for evaluation.

We select the yellow taxi data from January to March 2016 for a total of 91 days. It contains 34,498,859 records in all, including 19 fields such as *VendorID*, *pickup_datetime*, *dropff_datetime*, *passenger_count*, and *trip_distance*. We select several fields for cleaning and prediction, others are abandoned.

Figures 3 and 4 depict the statistical characteristics of NYC yellow taxi data. Figure 3a denotes the number of taxi pickups at each hour in one day. Figure 3b shows the number of taxi pickups in one week, we can see there is a day pattern in the wave. Another interesting distribution is shown in Figure 3c, which indicates that the strongest taxi demand is for short trips (shorter than 3 miles) or very long trips (longer than 25 miles).

lHeat maps in Figure 4 demonstrate the taxi demand distribution in the spatial dimension. Figure 4a–c represent distributions in Manhattan at 9:00, 14:00, and 22:00, respectively. From which we can see that strong demands (red colored) mostly covers acentric places at morning (Figure 4a) and mostly covers centric places at noon (Figure 4b), this may indicate residential areas dominate the acentric Manhattan while office buildings are mainly in centric Manhattan. and strong demands at night may show that some entertainment places are located in the south Manhattan (Figure 4c).

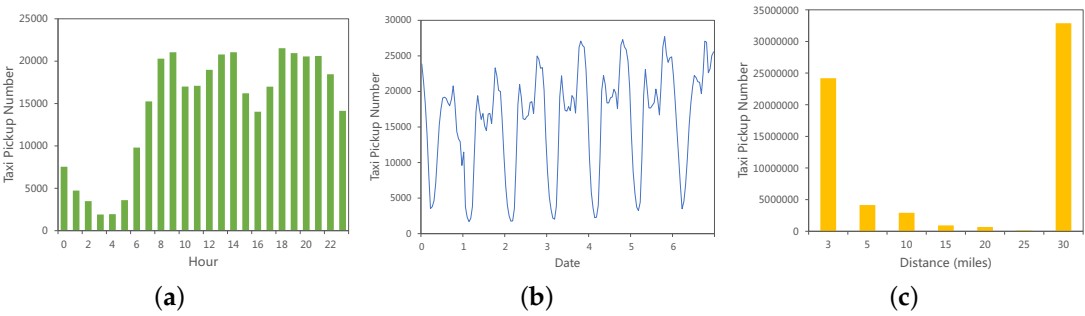

**Figure 3.** Distribution of New York City (NYC) Yellow Taxi Data. (**a**) The taxi pickup number in each hour of one day. (**b**) The taxi pickup number in each day of one week. (**c**) The grouped taxi pickup number at each distance range.

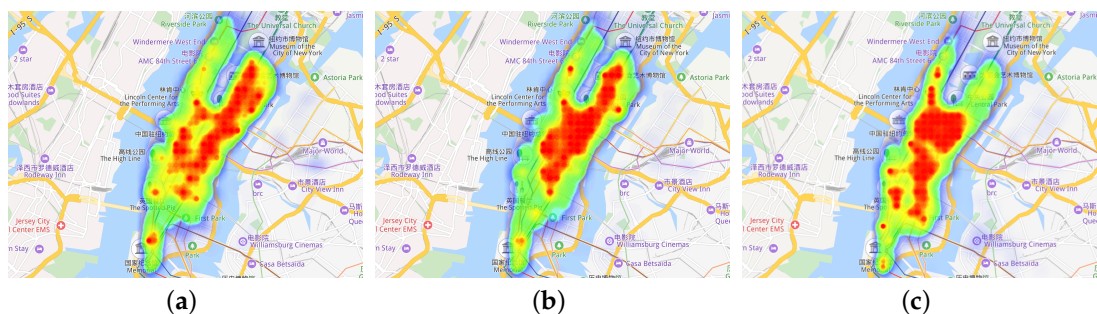

**Figure 4.** Heat maps of NYC Yellow Taxi Data in Manhattan. (**a**) Heat map at 9:00 of one day. (**b**) Heat map at 14:00 of one day. (**c**) Heat map at 22:00 of one day.

### 4.1.2. Data Cleaning

Data cleaning could clean out the noise data to improve the quality of the dataset. In addition to using the inspection mechanism provided by Python DataFrame, we further filtered the records with abnormal values via checking the following fields: *trip_duration*, *speed*, *fare_amount*, *pickup_coordinates* and *dropoff_coordinates*.

For the first three fields, we find the normal value range of each field, then remove records in the dataset with abnormal values in these fields. Take *fare_amount* for example, we analyze its values via a boxplot before cleaning up, the max value of *fare_amount* reaches an incredible $400,000. Further analysis found that 99.9% of fares are less than $100, so records with the 0.1% highest *fare_amount* are filtered out.

For the last two fields, we set the range of New York City is $(-74.255864, -73.701448)$ at longitude and $(40.495831, 40.915669)$ at latitude. If *pickup_coordinates* or *dropoff_coordinates* of one record is out of the range, this record will be filtered.

### 4.1.3. Data Featuring

In the existing works, external data are often used to make the features rich. In our work, we try to dig deep into the demand data, making the feature dimension more diverse.

**Feature 1: Time Features**

We select three time scales as part of external data: hour of day, day of week, month of year. As all they are finite discrete numbers, we use one-hot encoding to transform external data into binary vectors.

**Feature 2: Historical Records**

Historical values are the major feature for taxi demand prediction. To improve the prediction accuracy, we select $s$ historical values at previous weeks. As mentioned in Section 3.1.2, time interval amount of one day is $\left\lceil \frac{1440}{\phi} \right\rceil$, thus this amount of one week is $\tau = \left\lceil \frac{10080}{\phi} \right\rceil$. To predict taxi demand of $v_n$ at time interval $t$, the leveraged historical values are $\{D^{v_n}_{t-s*\tau}, D^{v_n}_{t-(s-1)*\tau}, ..., D^{v_n}_{t-\tau}\}$.

**Feature 3: Weighted Moving Average Predictions**

In addition to the historical values of the taxi demand, some transformations executed on historical data may also be taken to enrich the input features. In this study, the weighted moving average model (WMA) [55], which is a variant of the historical average model, is employed to generate prediction values as features. Assuming that the current time is $t$ and the current node is $v_n$, the number of historical values used to calculate the weighted moving average predicted value is $q$, then at time $t + 1$, the predicted formula is:

$$\tilde{D}^{v_n}_{t+1} = (q * D^{v_n}_t + (q - 1) * D^{v_n}_{t-1} + ... + 1 * D^{v_n}_{t-q+1}))/(q * (q + 1)/2). \tag{3}$$

In fact, there is another popular variant of historical average model named exponentially weighted moving average model (EWMA) [56]. We compare the prediction performance of WMA and EWMA on our taxi demand data, and the result shows that WMA has a higher accuracy, thus we choose WMA as feature here.

**Feature 4: Fourier Frequencies and Amplitudes**

Fourier transformation [57] converts a waveform with a repeating pattern from the time domain to the frequency domain and decomposes it into a series of sine waves with different frequencies and amplitudes. According to this principle, we consider the taxi demand wave in the 24-h period as a regular waveform, as shown in Figure 3b. Thus, it was converted into the frequency domain and decomposed into a series of sine waves. Frequencies and amplitudes of these sine waves are also used as features.

**Feature 5: Records of Destination Neighbors**

In addition to the OD relation being leveraged in the model to exploit greater guidance for taxi demand prediction in the O region, the historical taxi demand in D regions may also provide assistance as features. To make features more efficient, the 24-h period wave pattern is adopted again. We select the taxi dropoff number on the previous day to determine the features. For instance, for the time interval 9:00–9:20 on 6 March, the dropoff numbers in time intervals between 9:00 and 11:00 on 5 March are used to determine the features.

It must be noted that after obtaining these features, we combine the features of each grid at each time interval into a vector and then normalize it using a normal distribution, which further effectively reduce noise.

*4.2. Origin–Destination-Based Temporal Graph Attention Networks*

With the effectiveness of the OD-relation-based graph attention network, the prediction performance is improved obviously via our OD-TGAT model. In this subsection, the internal structure of OD-TGAT is described in detail.

4.2.1. Framework

Our framework is shown in Figure 5, which illustrates how to use the features of $s$ time intervals at time $t$ to predict the taxi demand for all nodes in the city graph at $t + 1$. There are four layers in the framework from bottom to top. The first layer at the bottom is the Transforming Layer whose goal is to generate origin–destination-based city graph and rich data features. The second layer is the

Spatial Layer, in which spatial correlations are captured by graph neural networks. The third layer is the temporal layer, in which recurrent neural networks are leveraged to exploit time dependencies. As the Output Layer, the top layer utilizes a fully-connected layer and a nonlinear activation function to perform the final prediction results generation.

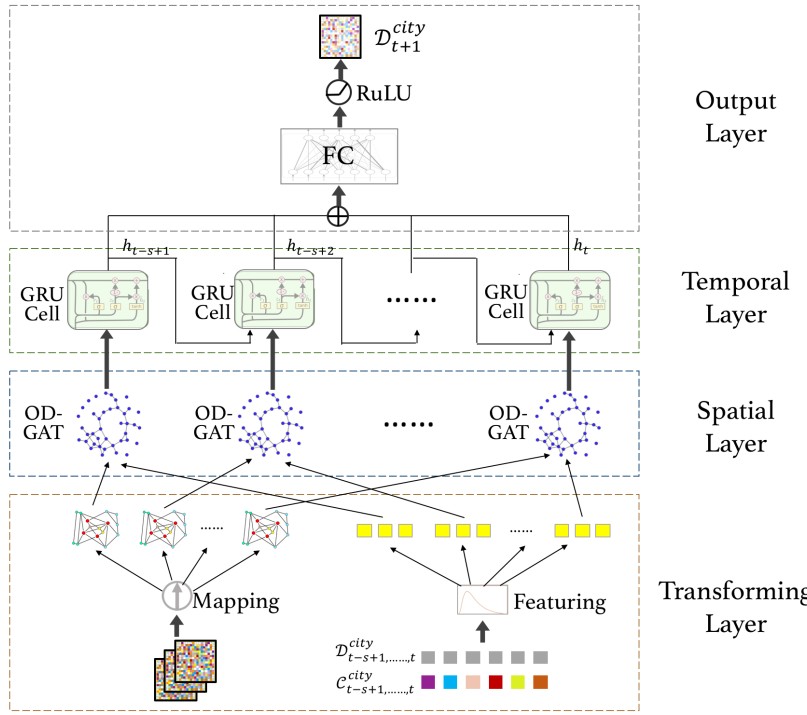

**Figure 5.** The Origin–Destination-based Temporal Graph Attention Network (OD-TGAT).

The details of the four layers are explained in the following subsections from the bottom layer to the top layer.

### 4.2.2. Data Transformation

Two tasks are performed in the transforming layer for each time interval $t - a (a \in [1 - s, ..., 0])$: mapping the grid map of the city to an OD-based city graph and extracting rich features from historical data and context data.

Firstly, given the grid map $M(\zeta)$ of Manhattan, New York, generated by city partitioning, as mentioned in Section 3.1.1, all taxi pickup and dropoff records are grouped into each grid, and grids with no record are filtered out. All remaining grids are mapped to nodes with indexes ordered from 0:

$$V = \{v_0, v_1, ..., v_N\}. \tag{4}$$

After that, we built the neighborhood for the nodes. For each origin node $v_i$, all taxi pickup records in the counterpart grid are listed. There is a dropoff grid in each of these records, and the nodes mapped from these dropoff grids constitute the neighborhood of $v_i$. Thus,

$$A_{ij} = \begin{cases} 1 & \text{if there is a trip from } v_i \text{ to } v_j, \\ 0 & \text{others} \end{cases} \tag{5}$$

The second task, data featuring, is elaborated in Section 4.1.3.

### 4.2.3. OD-GAT: Spatial Prediction Model

In the transforming layer, we convert the grid map into a city graph $\mathcal{G}_t$ and initializ the adjacency matrix $A$. In addition, features for grids at the current time interval are filled into nodes in $\mathcal{G}_t$. Node $v_i$ had a node feature $x_t$. The goal of the spatial prediction model, Origin–Destination-Graph AttenTion Networks (OD-GAT), is to help to update the features in each node under the influence of neighbors with OD relations.

Suppose node $v_i$ has $m$ neighbors. They have different distances from $v_i$ and share different taxi record numbers and time distributions with $v_i$, which results in different impacts on $v_i$. Therefore, it is necessary to quantize the influence of each neighbor. For node $v_i$ and one of its neighbors $v_j$, whose node features are $\vec{x}_i \in \mathbb{R}^F$ and $\vec{x}_j \in \mathbb{R}^F$, respectively, the attention coefficient of $v_j$ can be obtained using the following equation:

$$e_{ij} = a(W\vec{x}_i, W\vec{x}_j) \tag{6}$$

where $a$ is a mapping $\mathbb{R}^F \times \mathbb{R}^F \to \mathbb{R}$ which is implemented by a single-layer feed-forward neural network. $W \in \mathbb{R}^{F \times F}$ is a weight matrix. After normalization, the attention weight of $v_j$ is

$$\alpha_{ij} = \text{softmax}_j(e_{i,j}) = \frac{\exp(e_{ij})}{\sum_{k \in \mathcal{N}_i} \exp(e_{ik})} \tag{7}$$

where $\mathcal{N}_i$ is the neighbor set of $v_i$, and softmax$(\cdot)$ is a common and widely used normalized function. The above process is also illustrated in Figure 6a. After all attention weights of neighbors in $\mathcal{N}_i$ have been obtained, the updated hidden state of $v_i$ is then expressed as

$$\vec{x}_i' = \sigma\left(\sum_{j \in \mathcal{N}_i} \alpha_{ij} W\vec{x}_j\right). \tag{8}$$

In addition, like most self-attention applications, the multi-head is also incorporated into the attention mechanism. The idea behind the multi-head is assembling multiple self-attentions in parallel and using the concatenation or the average of the parallel results as the updated hidden state. In this way, the fitting ability of the model could be improved. As shown in Figure 6b, the representation of the updated hidden state of $v_i$ is replaced by a multi-head structure:

$$\vec{x}_i' = \|_{k=1}^K \sigma\left(\sum_{j \in \mathcal{N}_i} \alpha_{ij} W\vec{x}_j\right) \quad \text{or} \quad \vec{x}_i' = \sigma\left(\frac{1}{K}\sum_{k=1}^K \sum_{j \in \mathcal{N}_i} \alpha_{ij} W\vec{x}_j\right). \tag{9}$$

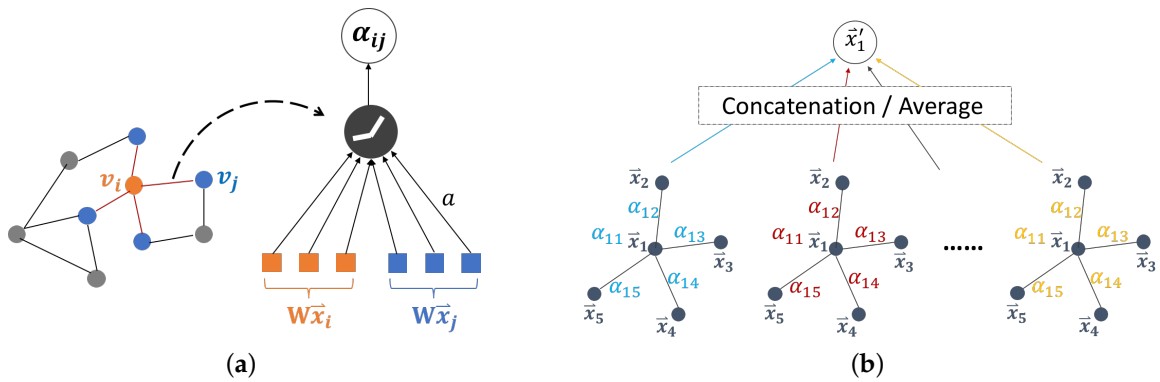

**Figure 6.** Inner Operations in OD-GAT. (**a**) Process to generate attention weight $\alpha_{ij}$. (**b**) Process to generate the new node feature $\vec{x}_i'$.

In the OD-GAT structure, we utilize the GAT structure, which could not only leverage features from neighborhood to help predict taxi demand on center nodes, but also allow neighbors to play

different roles in influencing the center node. Specifically, neighbors are defined based on OD relations, which could provide information from correlated non-adjacent areas to taxi demand prediction on the center node.

### 4.2.4. Temporal Prediction Model

The nature of taxi demand prediction is time series prediction, which guides us to select a proper model to process the sequential data. One of the most widely used models is the RNNs. Different from feed-forward neural networks, RNNs have a memory function which makes it predict outputs not only based on the current input but also depending on past ones. However, the initial version of RNNs suffers from gradient disappearance and gradient explosion in long-term prediction. In terms of these issues, series of variants can alleviate the gradient problems including the LSTM networks and the GRU. We choose the GRU for our sequential prediction, as it has a simpler structure and performs more efficiently than the LSTM.

More precisely, we adopt a GRU cell in our model. A GRU cell is actually a processing unit of the GRU. The GRU reads data from multiple time intervals at a time, and each GRU cell processes the data for each time interval. In our model, the data from each time interval needed to be handled by OD-GAT before inputting the temporal component. Therefore, we directly used the GRU cell to receive input from OD-GAT at each time interval.

As shown in Figure 7, at time interval $t$, the output hidden state of the GRU cell $h_t$ is determined by the following process based on two inputs: the results $x_t$ from the OD-GAT of the last layer and the hidden state generated by the GRU cell at the last time interval:

$$h_t = (1 - z_t) * h_{t-1} + z_t * \tilde{h}_t, \text{where} \begin{cases} \tilde{h}_t = tanh(W_{\tilde{h}} \cdot [r_t * h_{t-1}, x_t]) \\ z_t = \sigma(W_z \cdot [h_{t-1}, x_t]) \\ r_t = \sigma(W_r \cdot [h_{t-1}, x_t]). \end{cases} \tag{10}$$

Since the relation between the GRU cell and OD-GAT is one-to-one, when predicting the taxi demand result of one time interval, $s$ GRU cells and $s$ OD-GATs are needed, just as shown in Figure 5.

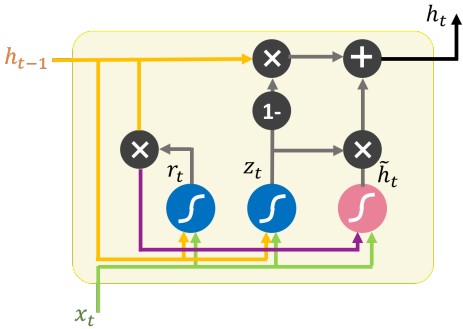

**Figure 7.** The structure of the Gated Recurrent Unit (GRU) cell where $x_t$ is from OD-GAT.

### 4.2.5. Output

Having generated $s$ hidden states from GRU cells in the third layer, the taxi demand of all nodes in $\mathcal{G}_t$ at time interval $t + 1$ is obtained after invoking a fully-connected layer:

$$\mathcal{D}_{t+1}^{\text{city}} = \text{ReLU}(W(\|_{r=0}^{s-1} h_{t-r}) + b). \tag{11}$$

## 5. Experiments

In this section, we compare our proposed model, OD-TGAT, with other state-of-the-art methods in terms of three metrics and analyze the performance of OD-TGAT with several parameters.

### 5.1. The Experimental Setup

The baselines, metrics, and parameter settings used in our evaluation are introduced in this subsection.

#### 5.1.1. Baselines

We compared our OD-TGAT with the following state-of-the-art approaches:

- **HA:** The prediction value of Historical Average (HA) of the next time interval is the average historical values of several previous relative time intervals. For example, if *time_step s* $= 3$, to predict the taxi demand at 21:00–21:20 on a given Sunday, the taxi demand values at 21:00–21:20 from the past three Sundays are taken to generate the result.
- **ARIMA:** ARIMA [58] is an representative time series prediction method, which is widely applied in statistics and econometrics. An ARIMA model may be considered as a "filter" which tries to separate the signal from noise and then the signal is extrapolated to make future forecasts.
- **LR:** Linear regression (LR) [59] is also a statistical model that is used to capture the relationship between a response and one or more explanatory variables. Based on this, it is employed to perform future predictions.
- **MLP:** Multilayer perceptron (MLP) [60] is an "original" version neural network. It has at least three layers of nodes—an input layer, a hidden layer, and an output layer—and it also uses backpropagation for training. Non-linear activation functions operate on nodes in the hidden layer and the output layer distinguish MLP from a linear approach.
- **Ridge:** Ridge regression [61] is actually a variant of linear regression. Different from the latter model, ridge regression leverages an L2-Regularization to reduce the sensitiveness to the noise from inputs.
- **RF:** Just as its name implies, Random Forest (RF) [62] build a forest consisting of decision trees via a random manner. It is able to improve the predicted accuracy observably with a small increasing computational cost.
- **XGBoost:** eXtreme Gradient Boosting (XGBoost), devised by Tianqi Chen, is a scalable tree boosting system which comprises many CART trees. It is so highly efficient, flexible, and portable that it has helped competitors beat champions in various worldwide competitions.
- **ST-ResNet:** ST-ResNet [63] is a popular model that has been proposed to use in the field of traffic prediction. It is a deep learning network that uses residual networks to model nearby and distant spatial correlations and three kinds of temporal dependencies.

#### 5.1.2. Metrics

Three metrics are used in this work to compare all methods: the *coefficient of determination (*$R^2$*)*, the *mean absolute error (*MAE*)* and the *root mean squared error (*RMSE*)*, given by

$$R^2 = 1 - \frac{\sum_{i=1}^{n}(y^{(i)} - \hat{y}^{(i)})^2}{\sum_{i=1}^{n}(y^{(i)} - \bar{y})^2} \tag{12}$$

$$\text{MAE} = \frac{1}{n}\sum_{i=1}^{n}|y^{(i)} - \hat{y}^{(i)}| \tag{13}$$

$$\text{RMSE} = \sqrt{\frac{1}{n}\sum_{i=1}^{n}(y^{(i)} - \hat{y}^{(i)})^2} \tag{14}$$

where $y^{(i)}$ and $\hat{y}^{(i)}$ are the $i$th ground truth value and predicted value of the taxi demand, respectively.

### 5.1.3. Experimental Setting

As mentioned in the Data Processing section, we use yellow taxi trip data from New York City as our dataset. We selected the time period from January 2016 to March 2016 of the data, which include 91 days in total. The first 80% (about 73 days) of days are used for training and the other 20% (about 18 days) are used for testing. In addition, in the spatial dimension, to make the evaluation more effective and efficient, we only use the Manhattan Borough in the experiments, which is the most bustling area in New York and the major area where the yellow taxis do business.

The model in this work is implemented in PyTorch (version 1.1) [64] which is based on Python. Moreover, the model is run on NVIDIA G40c, and each training process lasts for 3–5 h until the training loss is unable to be smaller in 50 epochs.

The grid size $\zeta$ is set to 200 m, and 423 non-empty grids are obtained for the three months. Each time interval $\phi$ is 20 min, so we are able to obtain the whole time sequence length, 6552, which is also the number of all time intervals. After that, the $s$ is set to 5, so at each time interval, features from the past 5 time intervals are used to forecast the next time interval. Furthermore, in terms of the network parameters, the GRU cell layer number is 1, the graph attention layer number is 2, and 32 heads are used in the graph attention layer.

### 5.2. Performance Comparison

### 5.2.1. Comparison with Baseline Models

Table 2 depicts the predictive performance of our OD-TGAT model and all other competing methods on the taxi demand testing dataset. We can observe that OD-TGAT achieves the lowest results among all approaches. More specifically, the OD-TGAT shows improvements of 5.5%, 9.3%, and 5.3% over the best baseline models in terms of MAE and RMSE, respectively. Besides, we can see that HA and ARIMA perform poorly (i.e., MAE values of 0.2267 and 0.2033, respectively, and RMSE values of 1.2214 and 0.9927). A possible reason for this is that they rely completely on historical demand values for prediction. In comparison, the better performance of regression methods (LR, Ridge, MLP, and XGBoost) reveals that they discover further correlations. In addition, ST-ResNet performs well, especially in terms of MAE and RMSE, which verifies its strong expression ability. However, ST-ResNet still performs worse than OD-TGAT, which attest the effectiveness of the graph networks in OD-TGAT.

**Table 2.** Performance Comparison with Baseline Models.

| Model | HA | ARIMA | LR | MLP | Ridge | RF | XGBoost | ST-ResNet | OD-TGAT |
|---|---|---|---|---|---|---|---|---|---|
| $R^2$ | 0.8163 | 0.9031 | 0.9264 | 0.9217 | 0.9272 | 0.9238 | 0.8638 | 0.9338 | 0.9849 |
| MAE | 0.2267 | 0.2033 | 0.1763 | 0.1801 | 0.1793 | 0.1922 | 0.1848 | 0.1628 | 0.1602 |
| RMSE | 1.2214 | 0.9927 | 0.9027 | 0.9072 | 0.8862 | 0.9191 | 0.8915 | 0.8374 | 0.7926 |

### 5.2.2. Performance at Different Time Moments

Further, we evaluate the prediction performance in each hour of one day and in each day of one week, as shown in Figure 8. Due to space limitations, only RMSE is demonstrated here. In Figure 8a, as we can see, the highest errors appear at the morning peak and night peak for all the three methods. This may be because demand values are relatively small at most hours of one day. After fitting the pattern, the model predicts a much more less value than the ground truth for morning and evening peak hours. Looking at Figure 8b, we can observe that the error is higher on weekends than on weekdays—almost 10% worse. This suggests that it is harder to predict taxi demand in the weekend. The reason for this may be that most people have more regular moving paths, such as commuting between home and the work place, on weekdays, while they have more choices during the weekend such as overtime working or short trips to other places. In both figures, the best performer is OD-TGAT,

followed by ST-ResNet, which shows that the relationships in spatial grids and temporal intervals are well captured by OD-TGAT.

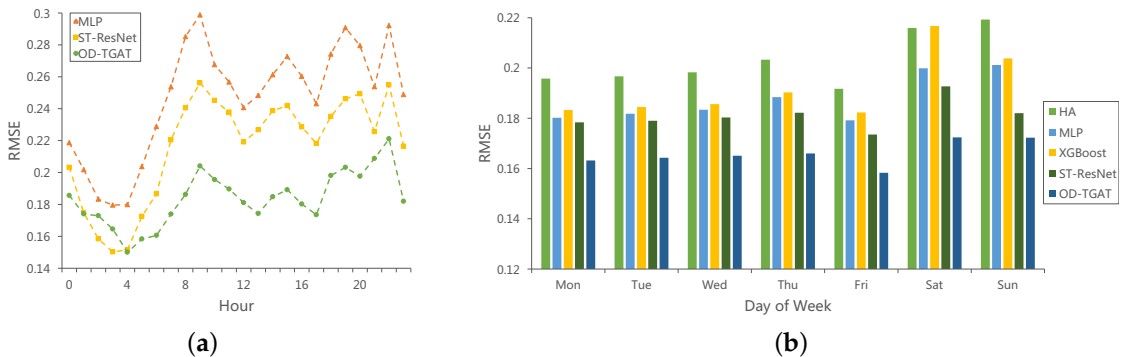

**Figure 8.** Performance on Different Time Scales. (**a**) Performance on different hours in one day. (**b**) Performance on different days in one week.

*5.3. Parameter Analysis*

We also conduct a parameter analysis for OD-TGAT. Three parameters are analyzed: *time_step*, *grid_size*, and *graph_attention_layer_number*. Figure 9a illustrates the prediction error of RMSE with respect to the *time_step*. We can see that when the *time_step* is 5, our method achieves the best performance. The decreasing trend in RMSE as the *time_step* increases shows the importance of longer temporal dependencies. The whole trend is stable, and the error increases when the *time_step* is greater than 5. In Figure 9b, we show the performance of our method with respect to both the size of the grid and the graph attention layer number. The best grid size is 300 m for all kinds of layer numbers. If the grid size is too small, there are too many single nodes with no neighbors to be benefit from. If the grid size is too large, the grid number decreases and the graph is nearly a complete graph, thus it is difficult to form non-equilibrium among nodes. In terms of the layer number, when it equals 3, the model achieves the best performance. It is also worth noting that when the grid size is large (more than 300), the bigger the layer number is, the lower the error of the model is. This may suggest that when all nodes in the graph become more homogenous, more attention layers mine deeper correlations, which is beneficial for the prediction accuracy.

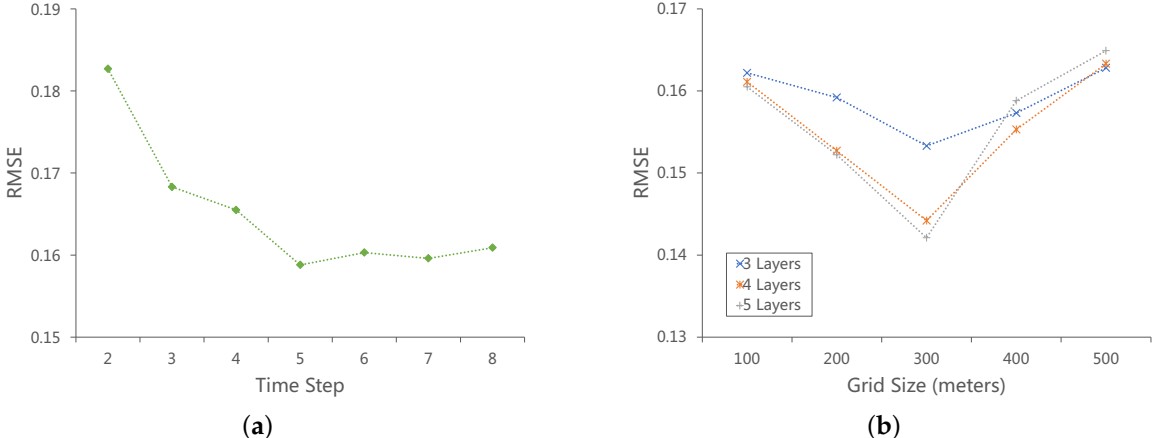

**Figure 9.** Performance on Parameters with Different Values. (**a**) Performance on different time step values. (**b**) Performance on different grid sizes and different attention layer numbers.

## 6. Conclusions

In this paper, we focus on a valuable and widely studied problem, taxi demand prediction, whose goal is to predict the taxi demand in future time intervals for a specified region. We argue that the existing works ignore the correlations between non-adjacent regions in the city and also do not utilize the power of the whole integrated city to achieve a more accurate forecast. In addition, they also lose sight of the fact that different spatial neighbors have different impacts on the center region.

To address these issues, we propose an Origin–Destination based Temporal Graph Attention Network (OD-TGAT) framework, which is the first employed model to graph networks for taxi demand prediction. In order to explore the relationships among non-adjacent regions, we propose an edge strategy called the OD Relation in OD-TGAT, which defines neighbors that have edges based on the OD relations of the same taxi. Further, an attention mechanism is adopted to compute the weight for each neighbor of each center region. In addition, the OD relation is used to enrich the input features, Moreover, various explorations on the origin taxi demand data are used to make the input features abundant.

How to better discover more effective correlations for higher quality prediction is still an open problem. In future work, we will further explore the relations between temporal and spatial data and leverage more external features for model improvement.

**Author Contributions:** Conceptualization, Y.X. and D.L.; methodology, Y.X. and D.L.; software, Y.X.; validation, Y.X.; formal analysis, Y.X.; investigation, D.L.; resources, D.L.; data curation, D.L.; writing–original draft preparation, Y.X.; writing–review and editing, D.L.; visualization, Y.X.; supervision, D.L.; project administration, D.L.; funding acquisition, D.L.

**Funding:** This research received no external funding.

**Conflicts of Interest:** The author declares no conflict of interest.

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
