# Peer review of "Incorporating Graph Attention and Recurrent Architectures for City-Wide Taxi Demand Prediction"

_ijgi, doi:10.3390/ijgi8090414_

Round 1

Reviewer 1 Report

This is an innovative article on predicting demand for taxi services. I believe it is appropriate for publication with a few changes to improve the clarity of the article.

I don’t follow the authors’ interpretation of Figure 5 – 5(a) is described as having more red area than 5(b), but visually it appears to have less.

Since I come from a more traditional statistics background, and I suspect many of the people interested in this article do as well, I would request that the authors spend more time on clearly defining the variables they are using to predict demand, particularly with regard to how they are selecting the time stamps. They list their features in Section 3.1.3. I’m not clear what Feature 1 is – is it the demand for particular days of the week and hours of the day, or simply an indicator for which day of the week and hour of day it is? If the latter, how is this variable used in prediction—wouldn’t it be the same for all connected grid cells?

Features 2, 3, and 5 seem to be aspects of historical demand – Feature 2 being the recent demand, 3 being historical demand, and 5 being demand at connected points on the previous days. First of all, I wonder if the same day of the previous week would be a better feature—for instance, I would expect taxi demand to be very different on Sunday and Monday? Secondly, why not use the demand from the previous day/week for the point being considered, rather than only its neighbors? Finally, since taxi demand is asymmetric (King, Peters, and Daus 2012), is Feature 5 using the demand for trips originating at the connected cell, or terminating there?

In Table 1, OD-TGAT is shown as having a lower R-squared than the other models tested. However, unlike the other metrics, a higher R-squared indicates a better model – is this a typographical error?

The authors note that the errors are highest in the morning and afternoon peaks, which they attribute to more complex demand patterns at that time. I wonder if the answer is simpler, and it is simply that you can be more wrong in absolute terms when the predicted value is larger? I would also encourage the authors to check the direction of the prediction errors during the peaks, to see if the model is significantly underpredicting demand at these times.

It seems the authors have only applied their model to Manhattan. I’d be curious how they handled trips that left Manhattan (many trips likely go to Brooklyn, Queens, the Bronx, etc.), and whether the model would be computationally feasible given a larger taxi market.

References

King, David A, Jonathan R Peters, and Matthew W Daus. 2012. “Taxicabs for Improved Urban Mobility: Are We Missing an Opportunity?” In TRBAM, 1–19.

Reviewer 2 Report

In this study, to predict the travel demand on the taxi mobility system, the authors use a customized RNN structure and derive different features from taxi trip records collected in New York. While there is need in the literature for data-driven mobility system studies using practical machine learning techniques, this manuscript needs to be completely re-written before reaching the standards of an academic report.

Abstract is badly written; especially Lines 9-13. Authors need to give an understandable overview instead of stating what they have done which requires details to be understandable.

When referring to an ANN structure in general use plural form, e.g. “RNNs have the ability to…”, or use the term “structure”.

Last two bullet points listed (Lines 76-81) make no sense as contributions of the study. They should not be listed there.

Due to high similarity of approaches mentioned in the section “Related works” I suggest, authors move it to after the “Introduction” section.

I suggest authors to take a look at these recent references closely relevant to their study and possibly review them in the text (regarding methodology and feature extraction form the data:

Ai, Y., Li, Z., Gan, M., Zhang, Y., Yu, D., Chen, W. and Ju, Y., 2019. A deep learning approach on short-term spatiotemporal distribution forecasting of dockless bike-sharing system. Neural Computing and Applications31(5), pp.1665-1677. Liu, L. and Chen, R.C., 2017. A novel passenger flow prediction model using deep learning methods. Transportation Research Part C: Emerging Technologies84, pp.74-91. Hamedmoghadam, H., Ramezani, M. and Saberi, M., 2019. Revealing latent characteristics of mobility networks with coarse-graining. Scientific reports9(1), p.7545.

What is #(.) used in Eq. (1)? Could you not use standard cardinality measure |.| notation?

I think I understand what authors mean by time_step h, but it needs to be changed and defined properly.

Completely rewrite the first paragraph of section 2.2., as it cannot be followed at all.

Matrix L is the symmetric normalized Laplacian, and is not the Laplacian matrix.

Why the green taxi data is not merged with yellow taxi data? The real demand for taxi is more accurate considering the both of these types.

There is no need for cumulative distribution in Fig. 4(c) to prove the point authors state in Lines 185&186. Also remove the statement.

I cannot stress enough how much redundant and irrelevant information is given throughout the manuscript. This paper can in half the number of pages, with only including important material and use of better writing.

Writing needs to be improved to a great extent. The text is full of errors and there is abundance of badly written sentences; everything from hard to read to unacceptable bad grammar. Authors should perform major revisions on the text before resubmitting the manuscript. Below I just listed a number of these problems in the first couple of pages:

Lines 29&30: bad sentence “Through simple and easy implemented, traditional approaches fail to…” Lines 38-41: very bad; no one can follow it. Line 42: “They does not…” Line 48: semicolon ends the sentence. Line 57: “… which is unable handled by CNN” Line 64: “… we summarize our contributions are” Line 96: “… which is benefit from simple and …” is just bad English. Lines 100&101: “The grid at the ith of longitude dimension and …” Lines 114&115: “The dividing is much easier….” makes no sense at all. Lines 132-134: remove the indent, C is already defined above and the rest of the sentence does not make sense. Last paragraph of Page 6 is very bad and does not convey much.

Reviewer 3 Report

This paper proposes a new graph method to predict taxi demand in urban areas. The topic is timely and interesting, and the approach is innovative. We live in an age where ridesharing systems are continuously being promoted as a way to overcome ever increasing congestion in urban developments. Predicting demand in a timely and efficient manner is a key for the success of these systems. Therefore, the current study could assist transport experts to come up with better and sustainable transport solutions for our communities.

However, there are some points in the manuscript that need to be attended and resolved before having this paper published. My comments are as follows,

The paper suffers from a poor English language and at some points undermines what authors want to convey. The article should be proofread by a professional proof-reader to enhance the quality of the paper. The “Related Work” section should come after “Introduction”. It is uncommon to see this section towards the end of the paper. The Related Work section should be enriched as well. The current study deals with ridesharing systems and in the abstract, it highlights the importance of demand prediction in reducing the imbalance of taxi scheduling between regions. There are plenty of studies in the literature that explore how ridesharing fleets should be managed to prevent the imbalance of taxis. These articles should be discussed and cited in the paper as well. Some of them are as follows:

Pavone,M.,S.L.Smith,E.Frazzoli,andD.Rus.2012.“RoboticLoadBalancingforMobility-on-Demand Systems.”TheInternationalJournalofRoboticsResearch31(7):839–854.

Spieser, K., K. Treleaven, R. Zhang, E. Frazzoli, D. Morton, and M. Pavone. 2014. “Toward a Systematic ApproachtotheDesignandEvaluationofAutomatedMobility-on-DemandSystems:ACaseStudy inSingapore.”229–245.SpringerInternationalPublishing.doi:10.1007/978-3-319-05990-7_20.

Farid Javanshour, Hussein Dia & Gordon Duncan (2018): Exploring the performance of autonomous mobility on-demand systems under demand uncertainty, Transportmetrica A: Transport Science, DOI: 10.1080/23249935.2018.1528485

Javanshour F., Dia H., Duncan G. (2019) Exploring System Characteristics of Autonomous Mobility On-Demand Systems Under Varying Travel Demand Patterns. In: Mine T., Fukuda A., Ishida S. (eds) Intelligent Transport Systems for Everyone’s Mobility. Springer, Singapore

Page4, Line116: st and et should be explained in the text. It is not clear what these parameters are. Page8, Line228: What is the reference for the method explained in “Feature3” ? It should be cited and discussed more. What is the “historical q value” ? Please elaborate on this a bit more. Page8, Line236: Similar to the previous comment, the method mentioned in “Feature4” should be explained with a reference. Page8, Line254: It says “With the effectiveness of OD relation based graph attention network, the prediction performance is improved obviously via our OD-TGAT model.”

The sentence is unclear. How did you figure out that the OD relation based graph attention network is effective? How did you realise that the prediction performance is improved OBVIOUSLY via OD-TGAT model? Please elaborate more on this and make it clear.

Round 2

Reviewer 2 Report

Still errors and hard-to-read sentences can be easily spotted in the text: Line 84: “demands data” Line 278: “… demands are distributed mostly in suburb at morning…” Line 303: the sentence makes no sense! Line 307: “an hour is 24 integers belonging to [0, 24)” Line 355: section should not be capitalized. Abstract is still hard to read. As I commented on it in the previous round of revision it needs to summarize the research in an understandable manner, rather than summarizing the methodology using ANN structure names! Last point listed as a “notable” contribution in the paper, still needs to be removed in my opinion. Authors can at least mention it elsewhere and simply say that they extract some extra features explaining the demand. There is abundance of studies using similar approaches to generate new features from data. #(.) is not a standard notation. Authors should define and use a cardinality function if they do not like |.|, and also they can use a slightly bigger set of || outside the curly brackets. Also, I suggest them to use a standard notation, as no one would be puzzled by correct usage of mathematical notations. There is only 23 integers within [0, 23). I don’t understand the sentence but I suggest using {1, 2, …, 24} or simply 1-24 instead. As said before, the manuscript is extremely long relative to what it is aimed to conduct. Authors should be concerned with the fact that a clean shorter version of this manuscript might attract way more readers, than a messy long manuscript. For example figure 6 is completely unnecessary, you just mention that in a sentence. Eq 5: improve it by writing simply as ‘1, if there is a trip from v_i to v_j’ and ‘0, otherwise’ Review and edit the References. There are many inconsistencies. For example both abbreviation and complete journal names are used which needs to be fixed according to journal’s required/preferred format.
